# Proposal of a New Parameter for Evaluating Muscle Mass in Footballers through Bioimpedance Analysis

**DOI:** 10.3390/biology11081182

**Published:** 2022-08-06

**Authors:** Matteo Levi Micheli, Roberto Cannataro, Massimo Gulisano, Gabriele Mascherini

**Affiliations:** 1Exercise Science Laboratory Applied to Medicine “Mario Marella”, Department of Experimental and Clinical Medicine, University of Florence, 50134 Florence, Italy; 2Department of Pharmacy, Health and Nutritional Sciences, University of Calabria, 87036 Rende, Italy

**Keywords:** Levi’s muscle index, LMI, lean mass

## Abstract

**Simple Summary:**

Body composition assessment in athletes is closely related to sports performance. In sports where alternating aerobic and anaerobic metabolism is required, muscle mass can affect the achievable result. Therefore, the assessment of muscle mass is a field of interest in the sports sciences. Bioimpedance analysis is a method for assessing body composition. Currently, bioimpedance evaluates muscle mass through predictive equations that could provide estimation errors. These errors could be reduced by an evaluation of the raw bioelectrical parameters. Therefore, in this study, we propose a new parameter for assessing the muscle mass in athletes.

**Abstract:**

The evaluation of muscle mass in athletes correlates with sports performance directly. Bioimpedance vector analysis is a growing method of assessing body composition in athletes because it is independent of predictive formulas containing variables such as body weight, ethnicity, age, and sex. The study aims to propose a new parameter (Levi’s Muscle Index, LMI) that evaluates muscle mass through raw bioelectrical data. A total of 664 male footballers underwent bioimpedance assessment during the regular season. LMI was correlated with body cell mass (BCM) and phase angle (PA) to establish efficacy. The footballers were 24.5 ± 5.8 years old, 180.7 ± 5.9 cm tall and weighed 76.3 ± 7.1 kg. The relationships were: LMI-BMI: r = 0.908, r^2^ = 0.824, *p* < 0.001; LMI-PA: r = 0.704, r^2^ = 0.495, *p* = 0.009 and PA-BCM: r = 0.491, r^2^ = 0.241, *p* < 0.001. The results obtained confirm that LMI could be considered a new parameter that provides reliable information to evaluate the muscle mass of athletes. Furthermore, the higher LMI-BCM relationship than PA-BCM demonstrates specificity for muscle mass evaluation in athletes regardless of body weight, ethnicity, age, and sex.

## 1. Introduction

Body composition characteristics play an important role in football performance; lower fat mass (FM) content and higher fat-free mass (FFM) are associated with better physical performance [1]. However, in high-level men’s sports, unlike high-level women’s sports, the evaluation of FFM and, in particular, muscle mass is particularly relevant for achieving higher sports performance [2,3,4].

One methodology for analyzing body composition in sport and, in particular, football is bioimpedance analysis (BIA) [5]. BIA provides direct measurements that allow non-invasive assessment of soft tissue evaluation in athletes [6] and can be performed (1) in the traditional way through predictive equations to quantify numerous body composition parameters or (2) through the impedance vector analysis (BIVA), which studies resistance (R) and reactance (XC) adapted to the height (H) of the subject, plotting them as a vector within a graph [7,8].

A previous study showed how football players could belong to a specific population, with FFM characterized by high body cell mass (BCM, traditional analysis) and phase angle (PA, vector analysis) [9,10]. These two parameters are directly related to football player performance [11]. They were also valuable in differentiating categories, as they were more pronounced in the elite than in the professional and semi-professional divisions [8]. BCM is the metabolically active component of FFM; the higher value in athletes may indicate better cell function [12] and endurance performance in soccer players [13]. PA, which is the arc tan of Xc/R [14], reflects the intracellular/extracellular water ratio, can be considered a parameter that informs cellular health [5,15] and is positively associated with the player’s aerobic power [16].

The evaluation of body composition using traditional BIA in athletes is undoubtedly useful, where the communicative value of the numerous parameters allows a quick and intuitive interpretation of the results. However, these results are obtained from predictive equations, which may not be population specific, compromising the evaluation’s reliability, especially in athlete samples [17].

Currently, the evaluation of muscle mass using BIA generally takes place starting from values obtained from the predictive formulas (in particular with the results of BCM and FFM), then carrying out a cross interpretation with the BIVA graph (in particular with PA) [5]. Therefore, the need arises to use a parameter that provides information on the state of muscle mass in athletes and can distinguish the training levels of players using only the raw bioelectrical values.

This study aims to propose a new parameter that does not derive from predictive equations but from parameters strictly related to BIVA and which, at the same time, can allow a rapid assessment of the state of muscle mass in athletes.

## 2. Materials and Methods

### 2.1. Study Design and Participants

A cross-sectional study was conducted on Italian football teams belonging to the first four categories:Serie A (the highest category of Italian professional football);Serie B (second highest category);Serie C (the lowest category of professional football);Serie D (national amateur championship).

Subjects were enrolled after receiving written informed consent. The study was carried out in conformity with the ethical standards laid down in the 1975 Declaration of Helsinki.

The following inclusion criteria were used: (1) male, Caucasian, and between the ages of 18 and 35, (2) registered with the sports club for the current season, (3) practiced football at a competitive level for at least ten years, (4) has had no injuries or surgery that could affect participation in physical activity in the previous three months, (5) not taking any medications.

### 2.2. Procedures

The recruitment and evaluation of the participants were conducted during the in-season phase (from September to May of a competitive football season). Therefore, the players showed their optimal body composition (i.e., the lowest FM and the highest FFM) [18]. After receiving the availability from the sports club and the team’s technical/medical staff, the morning before the last weekly training session before the championship match at the teams’ sports centers, all measurements were taken at rest by the same expert operator. In total, 664 male soccer players registered in four divisions of the Italian Soccer Federation (Federazione Italiana Giuoco Calcio) volunteered to participate in the study. Therefore, each subject entered the case study only once. Participants were divided into three groups according to their performance levels. The elite-level group consisted of 241 professional players participating in the first (Serie A) and second (Serie B) Italian divisions. The high-level group consisted of 223 professional players performing in the third (Serie C) Italian divisions. Finally, the medium-level group consisted of 200 semi-professional players from the fourth (Serie D) Italian divisions. The elite-level group performed an average of 22 training sessions and about six competitive games per month, the high-level group 20 training sessions and five competitive games per month, and the medium-level group 16 training sessions and four competitive games per month.

Weight was measured to the nearest 0.1 kg and height (H) to the nearest 0.5 cm (Seca GmbH & Co., Hamburg, Germany). Body-mass index (BMI) was then calculated as weight divided by H^2^ (kg/m^2^).

Bioelectrical impedance was measured using phase-sensitive plethysmography (BIA-101, Akern-RJL Systems, Florence, Italy). The device emitted an alternating sinusoidal electric current of 800 μA at a single operating frequency of 50 kHz. It was calibrated every morning using a calibration circuit procedure of known impedance (R = 380 Ohm, XC = 47 Ohm, 1% error) supplied by the manufacturer. Standard whole-body tetrapolar measurements were performed according to the manufacturer’s guidelines. The measurements were performed on the right side of the body with the subjects in a supine position with their arms and legs abducted [19].

BCM and fat mass (FM) (kg) was determined according to Kotler et al. [20], BCMI was defined as BCM/H^2^ (kg/m^2^), while PA was directly calculated from XC and R as the arctangent (XC/R; 180°/π).

The parameter we propose takes its name from the author who studied it first. It is called Levi’s Muscle Index, defined as LMI = (PA × H)/R.

### 2.3. Statistical Analysis

The data are expressed as mean ± SD. The variances’ equality and the sample distribution’s normality were analyzed using Levene’s test and a Shapiro–Wilk test, respectively. The comparison between the different football levels were performed using the one-way ANOVA test. The differences between groups were performed using an unpaired Student’s *t*-test for continuous variables. The relative effect sizes (ES) were calculated using Hedges’ g [21] to estimate the relevance of the differences analyzed. The ES were categorized as follows: <0.1 trivial;  ≥0.1 to  ≤0.3 small;  >0.3 to  ≤0.5 moderate; and  >0.5 large. Pearson’s correlation coefficient and coefficient of determination r^2^ calculated the statistical relationship between the new parameter LMI with BCM, BCMI, PA, FM, and between PA and BCM. Statistical analysis was performed using SPSS (Statistical Package for Social Science) (SPSS Inc., Chicago, IL, USA). The significance level was set as *p* < 0.05.

## 3. Results

Football players move from the lowest to the highest level by showing higher age, height, weight, and BMI (Table 1).

The same applies to bioimpedance parameters, both traditional (BCM and BCMI) and vector (R, XC, PA and LMI) (Table 2).

The relationships between LMI and body composition parameters BCM, BCMI, PA and FM are shown in Table 3 and in Figure 1. The relationship between PA and BCM was r = 0.491, r^2^ = 0.241, *p* < 0.001.

## 4. Discussion

The study aims to propose a new parameter for assessing muscle mass from bioimpedance analysis. LMI is designed to provide information through parameters closely related to the impedance vector and not derive from predictive equations that could skew the final result based on the test subject’s body weight, ethnicity, age, and gender.

The raw bioelectrical parameters in assessing FFM are R, XC, and PA. However, XC showed no sensitivity in evaluating football players. Therefore, only PA and R were used to generate the LMI formula. Finally, the LMI was normalized for the subject’s height to remove the influence of body length that would have affected the final conductivity.

The BIVA assessment allows us to detect a change in athletes’ hydration and/or muscle mass, regardless of body weight [22,23]. Furthermore, using the R/XC graph, it is observed that, with the same PA value, shorter vectors (therefore with lower R values) express greater muscle development [5]. Therefore, LMI consists of expressing the PA adjusted for the subjects’ height by dividing it by the R to make the phase variations independent from the hydration variations, thus associating them exclusively with the variations of the muscle component. As a result of these characteristics, the LMI could be used to assess athletes’ muscle mass without the traditional BIA or BIVA assessment. This is reasonable due to the ease of interpretation: higher values correspond to greater muscle mass.

A growing body of evidence indicates that PA can be applied to predict muscle quantity and strength, nutritional status, disease prognosis, and the likelihood of mortality [24,25]. A characteristic of PA is the rapid interpretation resulting from a simple number, unlike the other raw bioelectrical parameters (R and XC): LMI also seems to have this characteristic. This study suggests that higher LMI values correspond to higher footballer performance levels.

The results obtained in the present study confirm that the LMI is also a parameter capable of discriminating the different levels of football competition with a high ES, even higher than the PA. Furthermore, it has a strong direct relationship with the PA, which allows additional information to assess FFM in football players.

BCM assessment has a growing interest in assessing body composition by measuring impedance in sport and exercise. The role of BCM has been established in football for a long time [12,26]. Therefore, the fact that LMI shows an ES comparable to BCM and superior to BCMI in the discrimination of player membership levels and has an even stronger relationship than the PA denotes the reliability of this new parameter [27,28].

Two aspects that underline the sensitivity of LMI to the assessment of muscle mass in athletes are (1) the lack of association with FM, (2) the higher relationship between BCM and LMI (r = 0.908, r^2^ = 0.804) compared to BCM and PA (r = 0.491, r^2^ = 0.241).

The use of LMI could provide additional information in evaluating FFM due to the differences in BMI found among different levels of competition in the same sport. For example, relying only on BMI, which is higher in elite footballers, could lead to erroneous conclusions without a qualitative assessment of muscle-oriented FFM.

This study carried out a cross-sectional evaluation, demonstrating the usefulness of normalizing PA with R and, therefore, with a parameter related to the total body of water. A future direction of study could be to confirm the sensitivity of the LMI parameter in longitudinal studies in order to verify the achievement or maintenance of adequate body composition in athletes who are subject to daily variations in hydration status.

Bioimpedance analysis has numerous applications in the field of exercise and sports science. However, other fields, such as the medical and nutritional ones, could draw meaningful information from the new LMI parameter. Therefore, future studies could apply LMI in assessing growth status in the pediatric population [29] rather than in detecting nutritional status in underweight subjects [30] or the assessment of sarcopenia in the elderly population [31].

This study has some strengths. Firstly, LMI is to allow the evaluation of lean mass through raw bioelectrical parameters, therefore not linked by predictive equations. Secondly, LMI is a simple number, and its interpretation does not even need the BIVA plot. Third, the large sample size tested with the same instrumentation and the same operator allows for the reliability of the results.

The authors are aware of the limitations of this study. First, the LMI formula was obtained using foot-to-hand technology with a sampling rate of 50 kHz. Therefore, the results of this study cannot be extended to other impedance assessment solutions. Secondly, the subjects come from the same country; therefore, the differences between the levels of the categories may not be generalizable to all players in the world. Thirdly, it should be considered that the calculation of the accuracy of LMI in the evaluation of muscle mass was carried out through the values of BCM and PA, both measured with BIA. However, there is currently no reference method for measuring muscle mass with an indirect methodology. Generally, Dual-Energy X-ray Absorptiometry (DXA) is used as a reference methodology for body composition assessment; however, DXA does not evaluate muscle mass, but lean mass. Therefore, it is not sufficiently accurate enough; furthermore, there is the difficulty of transportation and the high costs. BCM and PA were chosen because (1) their relationship with the muscle mass associated with the performance of muscular strength is demonstrated [28]; (2) using foot-to-hand technology combined with predictive equations developed for athletes, BIA showed no differences from reference methods for FFM estimation [5]; (3) moreover, it allowed us to reach a large study sample that can provide statistically reliable information.

## 5. Conclusions

In summary, the new LMI parameter provides information about muscle mass in athletes. However, it should not replace other investigation methods such as laboratory analysis or field tests. Due to its simplicity of detection and interpretation, it could be used frequently during a competitive season.

New research directions could also allow its applicability to non-sporting contexts, where body composition assessment with particular attention to muscle mass provides information about people’s health.

## Figures and Tables

**Figure 1 biology-11-01182-f001:**
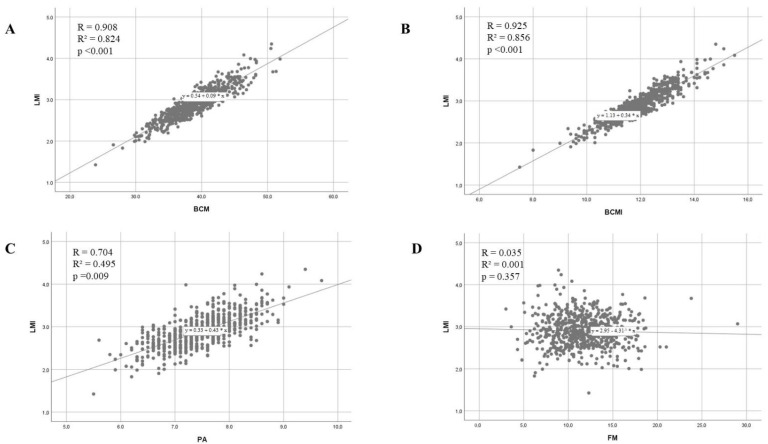
Scatterplots showing the relationships between LMI and (**A**) body cell mass (BCM), (**B**) body cell mass index (BCMI), (**C**) phase angle (PA), (**D**) fat mass (FM).

**Table 1 biology-11-01182-t001:** Age and anthropometric parameters of football players divided into three groups based on their level of performance. BMI = Body Mass Index; ES = Effect Size.

	Elite	High	Medium	F	ANOVA	Elite vs. High *p* Value (ES)	Elite vs. Medium *p* Value (ES)	High vs. Medium *p* Value (ES)
Age (yrs)	26.4 ± 6.5	24.4 ± 5.0	22.1 ± 5.1	32.7	<0.001	<0.001 (0.34)	<0.001 (0.74)	<0.001 (0.46)
Height (cm)	181.9 ± 6.1	180.8 ± 5.7	179.3 ± 5.4	11.1	<0.001	NS (0.19)	<0.001 (0.45)	<0.01 (0.27)
Weight (kg)	79.2 ± 6.7	76.0 ± 6.3	73.1 ± 6.9	45.5	<0.001	<0.001 (0.49)	<0.001 (0.89)	<0.001 (0.32)
BMI (kg/m^2^)	23.9 ± 1.4	23.2 ± 1.4	22.7 ± 1.7	35.1	<0.001	<0.001 (0.50)	<0.001 (0.78)	<0.01 (0.32)

**Table 2 biology-11-01182-t002:** Bioimpedance parameters of football players divided into three groups based on their level of performance. BCM = Body Cell Mass; BCMI = Body Cell Mass Index; R = Resistance; XC = Reactance; PA = Phase Angle; LMI = Levi’s Muscle Index; ES = Effect Size.

	Elite	High	Medium	F	ANOVA	Elite vs. High *p* Value (ES)	Elite vs. Medium *p* Value (ES)	High vs. Medium *p* Value (ES)
BCM (kg)	41.0 ± 3.6	38.7 ± 3.2	36.8 ± 3.6	76.6	<0.001	<0.001 (0.67)	<0.001 (1.17)	<0.001 (0.56)
BCMI (kg/m^2^)	12.4 ± 0.9	11.8 ± 0.8	11.5 ± 1.1	50.9	<0.001	<0.001 (0.70)	<0.001 (0.89)	<0.001 (0.31)
R (Ohm)	458.1 ± 38.4	470 ± 32.9	483.7 ± 47.5	22.2	<0.001	<0.001 (0.35)	<0.001 (0.59)	<0.001 (0.32)
XC (Ohm)	62.0 ± 6.7	61.3 ± 6.0	61.4 ± 7.0	0.8	0.423	NS (0.11)	NS (0.09)	NS (0.02)
PA (°)	7.7 ± 0.6	7.4 ± 0.5	7.2 ± 0.6	37.3	<0.001	<0.001 (0.54)	<0.001 (0.83)	<0.01 (0.36)
LMI (°∙cm∙Ω^−1^)	3.08 ± 0.35	2.87 ± 0.30	2.71 ± 0.36	65.7	<0.001	<0.001 (0.64)	<0.001 (1.04)	<0.001 (0.48)

**Table 3 biology-11-01182-t003:** Relationship between LMI and body composition parameters in the whole group of football players. LMI = Levi’s Muscle Index; BCM = Body Cell Mass; BCMI = Body Cell Mass Index; PA = Phase Angle; FM = Fat Mass.

LMI	r	r^2^	Β	CI 95%	t	*p*-Value
BCM (kg)	0.908	0.824	0.908	−0.66; −0.41	55.8	<0.001
BCMI (kg/m^2^)	0.925	0.856	0.925	−1.25; −0.99	62.8	<0.001
PA (°)	0.704	0.495	0.704	−0.58; −0.08	25.5	0.009
FM (kg)	0.035	0.001	−0.036	2.83; 3.06	−0.921	0.357

## Data Availability

Data can be obtained from Gabriele Mascherini at gabriele.mascherini@unifi.it.

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
