# Peer review of "Proposal of a New Parameter for Evaluating Muscle Mass in Footballers through Bioimpedance Analysis"

_biology, 2022, doi:10.3390/biology11081182_

Round 1

Reviewer 1 Report

Introduction

It is well written and structured. However, this paper needs more background. In fact, the manuscript is very poor in this section. It is a good starting point to place the reader. However, it would be helpful to elucidate in the first paragraph of introduction why you investigate about this topic? More references are necessary. In addition, is not clear you aim in the introduction.

I have not clear if all the actual literature was added? it would be interesting if the bibliography was updated.

Methodology

-          The information about the study design is not enough. Where is the register data? Where is the ethical committee?  

-          The methodology is not clearly explained and justified. Contact with the club, order of assessment, etc….

-          More information is needs about the groups

-          Can you create a section of materials? - Material, instruments…

-          The procedures are not clear. The procedure is not enough. Maybe a schematic representation of protocol will be needs to elucidate the work

Results

-          The statistical analysis is not correct. Can you show the effect sizes? I not understand the Table 1, why you perform an ANOVA and posteriorly a planned comparison post hoc? Why you perform this analysis?

-          Table and figure are bad? The name table is over the table and Figure under the table.

-          The results is clear. In addition, I think that the results will be explained in the discussion also.

Discussion and conclusion

Different aspect of conclusion should appear in the discussion section. The theoretical and practical implications of the research are vaguely mentioned at the end of paper. 

Author Response

The authors would like to thank the reviewer for appreciating our work and suggestions provided to improve our manuscript. Changes made to the manuscript are highlighted in red. Below are the answer to the reviewer’s comments.

Introduction

It is well written and structured. However, this paper needs more background. In fact, the manuscript is very poor in this section. It is a good starting point to place the reader.

However, it would be helpful to elucidate in the first paragraph of introduction why you investigate about this topic? More references are necessary. In addition, is not clear you aim in the introduction.

Answer: Thanks for the comment. The introduction section has been revised, four sentences and six new references have been added. In addition, the purpose of the study should now be more explicit at the end of this section.

I have not clear if all the actual literature was added? it would be interesting if the bibliography was updated.

Answer: Thanks for the comment; seven new references have been added. There are now 17 references in the introduction section.

Methodology

Reviewer The information about the study design is not enough. Where is the register data? Where is the ethical committee? 

Answer: Thanks for the comment; aspects relating to the ethical committee have already been addressed with the Editorial Committee.

Reviewer The methodology is not clearly explained and justified. Contact with the club, order of assessment, etc….

Answer: The evaluations were generally carried out on Saturday morning at rest before the training session (“the morning before the last weekly training session before the championship match”) during the period from September to May of the competitive season (“The recruitment and evaluation of the participants were conducted during the in-season phase.”).

Therefore, for greater clarity, the intervals of the months in which the assessments were carried out have been included.

Reviewer More information is needs about the groups

Answer: Thanks for the comment. The characteristics of the subjects are reported in the inclusion criteria and then in the variables used in the results. Other additional differences can be the number of training sessions and games per month. Then the sentence was added:

" The elite-level group performed an average of 22 training sessions and about six competitive games per month, the high-level group 20 training sessions and five competitive games per month, and the medium-level group 16 training sessions and four competitive games per month."

Reviewer Can you create a section of materials? - Material, instruments…

Answer: thank you for the advice.  The section “Data collection and Outcome measures” has been renamed “Procedures” and the name of instruments and the names of the instruments are in parentheses.

Reviewer The procedures are not clear. The procedure is not enough. Maybe a schematic representation of protocol will be needs to elucidate the work

Answer: The authors performed bioimpedance measurements on various soccer teams at different levels during the competitive season. Each subject had only one evaluation without evaluating changes over time. Therefore, each subject entered the case study only once. For this reason, this study was defined cross-sectional. In the authors' opinion, a schematic representation for a cross-sectional study is not necessary. However, a sentence was added at the beginning of the to clarify the procedures:

“After receiving the availability from the sports club and the team's technical/medical staff, the morning before the last weekly training session before the championship match at the teams' sports centers, all measurements were taken at rest by the same expert operator. In total, 664 male soccer players registered in four divisions of the Italian Soccer Federa-tion (Federazione Italiana Giuoco Calcio) volunteered to participate in the study. There-fore, each subject entered the case study only once.”

Results

Reviewer The statistical analysis is not correct. Can you show the effect sizes? I not understand the Table 1, why you perform an ANOVA and posteriorly a planned comparison post hoc? Why you perform this analysis?

Answer: ANOVA test results do not map out which groups are different from other groups. Therefore, after ANOVA a t-test were performed. Effect Size has already been shown in table 1 and 2. For clarity, the p-value of the t-test is also entered in Tables 1 and 2 along with the ES.

Reviewer Table and figure are bad? The name table is over the table and Figure under the table.

Answer: I apologize for the inconvenience, it was a pagination error. In the new version, the authors will pay attention to propose a more understandable version of the manuscript in relation to legends-tables-figures. The new version of the manuscript proposes a figure/table on a single page.

Reviewer The results is clear. In addition, I think that the results will be explained in the discussion also.

Answer: Thanks for the comment.

The authors believe they have commented on the results of the discussion with the exception of those relating to table 1. Therefore a sentence has been added:

“The use of LMI could provide additional information in evaluating FFM due to the differences in BMI found among different levels of competition in the same sport. For example, relying only on BMI, found higher in elite footballers, could lead to erroneous conclusions without a qualitative assessment of muscle-oriented FFM.”

For clarity, I report those already discussed:

  • About XC results in table 2: “The raw bioelectrical parameters in assessing FFM are R, XC, and PA. However, XC showed no sensitivity in evaluating football players. Therefore, only PA and R were used to generate the LMI formula.”
  • About LMI in table 2: “The results obtained in the present study confirm that the LMI is also a parameter capable of discriminating the different levels of football competition with a high ES, even higher than the PA. Furthermore, it has a strong direct relationship with the PA, which allows additional information to assess FFM in football players.”
  • LMI in comparison to PA, BCM and BCMI in table 2: “Therefore, the fact that LMI shows an ES comparable to BCM and superior to BCMI in the discrimination of player membership levels and has an even stronger relationship than the PA denotes the reliability of this new parameter”
  • r-value of LMI (table 3, figure 1): “Two aspects that underline the sensitivity of LMI to the assessment of muscle mass in athletes are 1) the lack of association with FM, 2) the higher relationship between BCM and LMI (r = 0.908, r2 = 0.804) compared to BCM and PA (r = 0.491, r2 = 0.241).”

Discussion and conclusion

Reviewer Different aspect of conclusion should appear in the discussion section. The theoretical and practical implications of the research are vaguely mentioned at the end of paper.

Answer: Thank you for your advice. Some aspects of the conclusions were reflected in the discussion.

“As a result of these characteristics, the LMI could be used to assess athletes' muscle mass without the traditional BIA or BIVA assessment. This is reasonable due to the ease of interpretation: higher values correspond to greater muscle mass.”

The conclusions have been reworked.

“In summary, the new LMI parameter provides information about muscle mass in athletes.  However, it should not replace other investigation methods such as laboratory analysis or field tests. Due to its simplicity of detection and interpretation, it could be used frequently during a competitive season.

New research directions could also allow its applicability to non-sporting contexts, where body composition assessment, with particular attention to muscle mass, provides information about people's health.”

Reviewer 2 Report

This manuscript proposes a new parameter for assessing muscle mass in footballers with different performance levels. Although this may have potential, the accuracy of the LMI has been compared to other components of BIA; the latter has multiple limitations. Determining the accuracy of LMI would have benefited by being compared to a gold standard method, and not to other components of BIA.

Additionally, it is not clear why LMI is useful and what would be its implications. Does it help athletes upgrade to a higher level but looking to increase muscle mass as a result? It is also unclear to me what was the point in differentiating different levels of footballers in relation to LMI. I find the objectives and the results to be unclear.

The content of the manuscript is limited, and it would benefit by including further information that help the readers have a clear understanding of the study conducted. As it stands, this paper is not suitable for publication. I have added below some further comments that may be of help.

Major comments:

Introduction is very limited and significant information is missing here. You are attempting to publish in “Biology”; hence many readers are unfamiliar with your topic and several terms/statements need to be clarified:

         Line 33: Explain further how body composition plays a role in football performance (e.g more muscle mass à better performance?)

        Lines 42 & 43: BCM and phase angle must be defined/explained.

        Lines 56 & 57: Discuss BIVA in additional to previous research that led to using this parameter.

Lines 62: For those who don’t understand the Football system in Italy, it is important to explain the 4 series here (it was only explained later).

Lines 83-84: Please indicate the equipment used to measure weight and height.

Lines 96-97: Please explain further or add previous references explaining the parameter.

Table 2: Please explain how you calculated BCMI.

Discussion: It is not clear how does significant difference in age and BMI affect your results?  It is also unclear here how categorizing players have helped looked at the accuracy of LMI.

Minor comments

Line 16:  “could” reads better than “should” be reduced.

Line 36: BIA is Bio impendence analysis

Line 61: was conducted “on”

Author Response

The authors would like to thank the reviewer for appreciating our work and suggestions provided to improve our manuscript. Changes made to the manuscript are highlighted in red. Below are the answer to the reviewer’s comments.

Reviewer This manuscript proposes a new parameter for assessing muscle mass in footballers with different performance levels. Although this may have potential, the accuracy of the LMI has been compared to other components of BIA; the latter has multiple limitations. Determining the accuracy of LMI would have benefited by being compared to a gold standard method, and not to other components of BIA.

Answer:  Thanks for the comment. The authors agree with the reviewer; however, there is currently no absolute reference method for muscle mass with an indirect body composition assessment methodology. Generally, DXA is used as a reference methodology; however, DXA does not evaluate muscle mass but lean mass. Therefore, it is not sufficiently accurate in addition to the difficulty of transportation and high costs.

BCM (traditional BIA) and PA (vector BIA) parameters were chosen for comparison because the literature has amply demonstrated their relationship with the muscle component associated with muscle strength performance (Custódio Martins P, de Lima TR, Silva AM, Santos Silva DA. Association of phase angle with muscle strength and aerobic fitness in different populations: A systematic review. Nutrition. 2022 Jan;93:111489. doi: 10.1016/j.nut.2021.111489). This allowed us to reach a large study sample which is, therefore, able to provide statistically reliable information.

Furthermore, using foot-to-hand technology combined with predictive equations developed for athletes, the BIA showed no differences with the reference methods for estimating FFM (Campa F, Gobbo LA, Stagi S, Cyrino LT, Toselli S, Marini E, Coratella G. Bioelectrical impedance analysis versus reference methods in the assessment of body composition in athletes. Eur J Appl Physiol. 2022 Mar;122(3):561-589. doi: 10.1007/s00421-021-04879-y).

Reviewer Additionally, it is not clear why LMI is useful and what would be its implications. Does it help athletes upgrade to a higher level but looking to increase muscle mass as a result? It is also unclear to me what was the point in differentiating different levels of footballers in relation to LMI. I find the objectives and the results to be unclear.

The content of the manuscript is limited, and it would benefit by including further information that help the readers have a clear understanding of the study conducted. As it stands, this paper is not suitable for publication. I have added below some further comments that may be of help.

Answer: Thanks for the comment; the manuscript has changed to make it more understandable, especially in the introduction section. Four sentences and six new references have been added to provide more background. In addition, the purpose of the study should now be more explicit at the end of this section.

Major comments:

Reviewer Introduction is very limited and significant information is missing here. You are attempting to publish in “Biology”; hence many readers are unfamiliar with your topic and several terms/statements need to be clarified:

Line 33: Explain further how body composition plays a role in football performance (e.g more muscle mass à better performance?)

Answer: Thanks for the comment; the introduction section has been revised, highlighting the importance of assessing fat-free mass and the need for a more muscle-oriented parameter. In addition, new references have been added.

Reviewer Lines 42 & 43: BCM and phase angle must be defined/explained.

Answer: BCM and PA have been better defined, relating their value to sports performance.

Reviewer Lines 56 & 57: Discuss BIVA in additional to previous research that led to using this parameter.

Answer: Thanks for the comment. The paragraph was revised because it could be misunderstood, in fact the sentence changing the reviewer's request had another meaning. Therefore now the paragraph before the purpose of the study is:

“Currently, the evaluation of muscle mass using BIA generally takes place starting from values obtained from the predictive formulas (in particular with the results of BCM and FFM), then carrying out a cross interpretation with the BIVA graph (in particular with PA). Therefore, the need arises to use a parameter that provides information on the state of muscle mass in athletes and can distinguish the training levels of players using only the raw bioelectrical values.”

Reviewer Lines 62: For those who don’t understand the Football system in Italy, it is important to explain the 4 series here (it was only explained later).

Answer: an explanation has been added. Now the sentence is:

“A cross-sectional study was conducted on Italian football teams belonging to the first four categories:

  • Serie A (the highest category of Italian professional football),
  • Serie B (second highest category),
  • Serie C (the lowest category of professional football),
  • Serie D (national amateur championship).”

Reviewer Lines 83-84: Please indicate the equipment used to measure weight and height.

Answer: Thank you for the comment. The equipment has been added.

Reviewer Lines 96-97: Please explain further or add previous references explaining the parameter.

Answer: Thanks for the comment; a reference was added in the introduction section where we were asked to define the PA in advance to the methods section.

Reviewer Table 2: Please explain how you calculated BCMI.

Answer: BCMI was defined together with the other variables in the methods section as BCMI was defined as BCM / H² (kg / m²) now at line 114.

Discussion:

Reviewer It is not clear how does significant difference in age and BMI affect your results?  It is also unclear here how categorizing players have helped looked at the accuracy of LMI.

Answer: Thank you for being able to clarify this aspect. Table 1 demonstrates that elite athletes have different characteristics; even parameters that might provide little information, such as age and BMI, actually provide additional information on the peculiarities of individuals participating in different levels of the same sport. For example, if age could provide information on sports experience, the aspects related to BMI appear more relevant. To emphasize this, a sentence has been added in the discussion section:

“The use of LMI could provide additional information in evaluating FFM due to the differences in BMI found among different levels of competition in the same sport. For example, relying only on BMI, found higher in elite footballers, could lead to erroneous conclusions without a qualitative assessment of muscle-oriented FFM.”

Minor comments

Reviewer Line 16:  “could” reads better than “should” be reduced.

Answer: Thank you for the suggestion. The change was made accordingly.

Reviewer Line 36: BIA is Bio impendence analysis

Answer: Thank you for the suggestion. The change was made accordingly.

Reviewer Line 61: was conducted “on”

Answer: Thank you for the suggestion. The change was made accordingly.

Reviewer 3 Report

First of all, I would like to congratulate the authors for their research and effort.

In general, the article can be published after a few minor revisions. I have listed the revisions below.

-In the introduction, it is necessary for the readers to understand it more clearly and to deal with the subject in more detail.

-You can include a protocol process chart for better understanding of the research design.

-Please indicate the ethics committee acceptance number.

-There are shifts in figures in the tables, please correct.

- Please specify the confidence intervals of the r values.

-Please improve the conclusion and discussion section.

Author Response

First of all, I would like to congratulate the authors for their research and effort.

In general, the article can be published after a few minor revisions. I have listed the revisions below.

Answer: The authors would like to thank the reviewer for appreciating our work and suggestions provided to improve our manuscript. Changes made to the manuscript are highlighted in red. Below are the answer to the reviewer’s comments.

Reviewer In the introduction, it is necessary for the readers to understand it more clearly and to deal with the subject in more detail.

Answer: Thanks for the comment. The introduction section has been revised, four sentences and six new references have been added. In addition, the purpose of the study should now be more explicit at the end of this section.

Reviewer You can include a protocol process chart for better understanding of the research design.

Answer: The authors performed bio impedance measurements on various soccer teams at different levels during the competitive season. Each subject had only one evaluation without evaluating changes over time. Therefore, each subject entered the case study only once. For this reason, this study was defined cross-sectional. In the authors' opinion, a schematic representation for a cross-sectional study is not necessary.

The evaluations were generally carried out on Saturday morning at rest before the training session (“the morning before the last weekly training session before the championship match”) during the period from September to May of the competitive season (“The recruitment and evaluation of the participants were conducted during the in-season phase.”).

Therefore, for greater clarity, the intervals of the months in which the assessments were carried out have been included. In addition, a sentence was added to clarify the procedures. Now the beginning of Procedures is:

“The recruitment and evaluation of the participants were conducted during the in-season phase (from September to May of a competitive football season). Therefore, the players showed their optimal body composition (i.e., the lowest FM and the highest FFM) [18]. After receiving the availability from the sports club and the team's technical/medical staff, the morning before the last weekly training session before the championship match at the teams' sports centers, all measurements were taken at rest by the same expert operator. In total, 664 male soccer players registered in four divisions of the Italian Soccer Federation (Federazione Italiana Giuoco Calcio) volunteered to participate in the study. There-fore, each subject entered the case study only once.”

Reviewer Please indicate the ethics committee acceptance number.

Answer: Thanks for the comment, aspects relating to the ethics committee have already been addressed with the Editorial Committee.

Reviewer There are shifts in figures in the tables, please correct.

Answer: I apologize for the inconvenience, it was a pagination error. In the new version, the authors will pay attention to propose a more understandable version of the manuscript in relation to legends-tables-figures. The new version of the manuscript proposes a figure/table on a single page.

Reviewer Please specify the confidence intervals of the r values.

Answer: CI 95% has been added in table 3.

Reviewer Please improve the conclusion and discussion section.

Answer: Thank you for the advice.

Two sentence of discussion has been added:

  1. “As a result of these characteristics, the LMI could be used to assess athletes' muscle mass without the traditional BIA or BIVA assessment. This is reasonable due to the ease of interpretation: higher values correspond to greater muscle mass.”
  2. “The use of LMI could provide additional information in evaluating FFM due to the differences in BMI found among different levels of competition in the same sport. For example, relying only on BMI, found higher in elite footballers, could lead to erroneous conclusions without a qualitative assessment of muscle-oriented FFM.”

Now le conclusion is:

“In summary, the new LMI parameter provides information about muscle mass in athletes.  However, it should not replace other investigation methods such as laboratory analysis or field tests. Due to its simplicity of detection and interpretation, it could be used frequently during a competitive season.

New research directions could also allow its applicability to non-sporting contexts, where body composition assessment, with particular attention to muscle mass, provides information about people's health.”

Round 2

Reviewer 1 Report

Introduction

It is well written and structured. However, this paper needs more background. In fact, the manuscript is very poor in this section. It is a good starting point to place the reader. However, it would be helpful to elucidate in the first paragraph of introduction why you investigate about this topic? More references are necessary. In addition, is not clear you aim in the introduction.

I have not clear if all the actual literature was added? it would be interesting if the bibliography was updated.

Methodology

-       The information about the study design is not enough. Where is the register data? Where is the ethical committee?  

-       The methodology is not clearly explained and justified. Contact with the club, order of assessment, etc….

-       More information is needs about the groups

-       Can you create a section of materials? - Material, instruments…

-       The procedures are not clear. The procedure is not enough. Maybe a schematic representation of protocol will be needs to elucidate the work

Results

-       The statistical analysis is not correct. Can you show the effect sizes? I not understand the Table 1, why you perform an ANOVA and posteriorly a planned comparison post hoc? Why you perform this analysis?

-       Table and figure are bad? The name table is over the table and Figure under the table.

-       The results is clear. In addition, I think that the results will be explained in the discussion also.

Discussion and conclusion

Different aspect of conclusion should appear in the discussion section. The theoretical and practical implications of the research are vaguely mentioned at the end of paper. 

Author Response

Dear Reviewer 1,
Thank you for the work and requests for changes to improve our manuscript.

In the second review round, the authors noted an increased appreciation of the manuscript by responding positively to all questions on the review report form.

However, the same comments from the first round of review appear in the "Comments and Suggestions for Authors." The authors have already replied and edited the manuscript accordingly. Given this, the authors ask, to Reviewer and Editor, if all the answers previously given were not exhaustive or if there is a documentation error. Therefore, there are no requests for changes or clarifications required.

Reviewer 2 Report

Thank you for addressing most comments and clarifying several points. I still, however, have reservations over the accuracy of the use of BCM and PA parameters. Please include a statement in the limitations section discussing the drawbacks of using these parameters in relation to the accuracy of LMI.

Author Response

Dear Reviewer 2,
The authors are grateful for the work done to improve the manuscript.

Reviewer: Thanks for addressing most of the comments and clarifying several points. However, I still have reservations about the accuracy of using the BCM and PA parameters. Please include a statement in the limitations section discussing the disadvantages of using these parameters in relation to the accuracy of the LMI.

Authors: Thanks for the comment. The authors agree to include a third study limitation in order to highlight the issue raised by the reviewer to the reader. Therefore, a new sentence has been inserted at the end of the discussion section:

"Thirdly, it should be considered that the calculation of the accuracy of LMI in the evaluation of muscle mass was carried out through the values ​​of BCM and PA, both measured with BIA. However, there is currently no reference method for measuring muscle mass with an indirect methodology. Generally, Dual-Energy X-ray Absorptiometry (DXA) is used as a reference methodology for body composition assessment; however, DXA does not evaluate muscle mass but lean mass. Therefore, it is not sufficiently accurate enough, besides the difficulty of transport and the high costs. BCM and PA were chosen because

  1. their relationship with the muscle mass associated with the performance of muscular strength is demonstrated [28];
  2. using foot-to-hand technology combined with predictive equations developed for athletes, BIA showed no differences with reference methods for FFM estimation [5];
  3. Moreover, it allowed us to reach a large study sample that can provide statistically reliable information."